# Loco-Regional Treatment of the Primary Tumor in De Novo Metastatic Breast Cancer Patients Undergoing Front-Line Chemotherapy

**DOI:** 10.3390/cancers14246237

**Published:** 2022-12-17

**Authors:** Corrado Tinterri, Andrea Sagona, Erika Barbieri, Simone Di Maria Grimaldi, Flavia Jacobs, Alberto Zambelli, Rubina Manuela Trimboli, Daniela Bernardi, Valeriano Vinci, Damiano Gentile

**Affiliations:** 1Breast Unit, IRCCS Humanitas Research Hospital, Via Manzoni 56, Rozzano, 20089 Milan, Italy; 2Department of Biomedical Sciences, Humanitas University, Via Rita Levi Montalcini 4, Pieve Emanuele, 20090 Milan, Italy; 3Medical Oncology and Hematology Unit, IRCCS Humanitas Research Hospital, Via Manzoni 56, Rozzano, 20089 Milan, Italy

**Keywords:** metastasis, breast cancer, loco-regional treatment, surgery, chemotherapy

## Abstract

**Simple Summary:**

Approximately 8% of breast cancers are diagnosed with synchronous distant metastasis at initial diagnosis, a situation known as de novo metastatic disease. Despite advances in the systemic treatment of metastatic disease, the optimal management of de novo metastatic breast cancer has been a matter of debate over the past few decades. Several studies have investigated the role of loco-regional treatment of the primary tumor and whether it was associated with better oncological outcomes. However, the results of these trials have been highly heterogeneous and inconsistent, leaving this question unresolved. In this retrospective study, we aimed to investigate the characteristics, treatment, and long-term outcomes of a cohort of consecutive patients with de novo metastatic disease who received loco-regional treatment after front-line chemotherapy. The results of this analysis may help identify interesting differences among de novo metastatic breast cancer patients that could help clarify the management of this controversial subgroup of patients.

**Abstract:**

Background: Loco-regional therapy (LRT) in de novo metastatic breast cancer (MBC) has been investigated in several clinical trials, with heterogeneous and conflicting results. Methods: We conducted a retrospective study of de novo MBC patients treated with front-line chemotherapy (FLC) followed by LRT of the primary tumor. Our aims were to evaluate the characteristics, treatment, and oncological outcomes in terms of progression-free survival (PFS), distant progression-free survival (DPFS), and overall survival (OS) of de novo MBC. We also investigated possible subgroups of patients with better outcomes according to menopausal status, biological sub-type, location, number of metastases, and radiologic complete response after FLC. Results: We included 61 patients in the study. After a median follow-up of 55 months, disease progression occurred in 60.7% of patients and 49.2% died. There were no significant differences in PFS, DPFS, and OS between different subgroups of de novo MBC patients. A trend toward better PFS and DPFS was observed in triple-positive tumors, without a statistically significant difference in OS. Conclusions: No specific subgroup of de novo MBC patients showed a statistically significant survival advantage after FLC followed by LRT of the primary tumor.

## 1. Introduction

Approximately 3–8% of breast cancer patients experience synchronous distant metastases at first presentation [1,2,3]. Metastatic breast cancer (MBC) is considered an incurable disease and, according to international guidelines published by the National Comprehensive Cancer Network (NCCN), the standard treatment for de novo MBC is systemic therapy without resection of the primary tumor [4]. Patients are classified as having de novo MBC if they present with an advanced disease without having a previous diagnosis at an earlier stage of breast cancer; this excludes patients who have received prior therapy and relapsed [5]. Recent advances in the systemic treatment landscape have considerably improved disease control in the metastatic setting and led to better oncological outcomes, particularly in human epidermal growth factor receptor 2-positive (HER2+) and hormone receptor-positive (HR+) breast cancer [6,7,8]. Surgery on the primary site is usually recommended in MBC patients for palliative reasons to alleviate symptoms and improve quality of life [4,9]. The role of loco-regional treatment (LRT) with surgery in patients with MBC at first presentation is controversial. It is certainly true that the resection of the primary tumor improves local disease control by slowing down the progression of breast cancer; however, its effect on the prognosis of MBC patients remains unclear [10,11]. Empirical evidence suggests that surgery on the primary tumor may promote metastatic spread [12]; however, many retrospective studies [10,13,14,15] and meta-analyses [16,17] showed a beneficial effect of LRT in limited subsets of MBC patients. These retrospective analyses vary in terms of patient populations, timing, and type of surgery; moreover, they have been highly criticized for their intrinsic selection bias and limited ability to control potential confounding factors. In fact, MBC patients who underwent LRT tended to be younger and with limited metastatic disease compared with patients who underwent systemic therapy. To overcome these limitations, two prospective randomized controlled clinical trials have been published, albeit with different designs and contrasting results [18,19]. Given these conflicting results, we conducted a retrospective analysis of consecutive de novo MBC patients who underwent front-line chemotherapy (FLC) followed by LRT at our Institution. This study aimed to evaluate the characteristics, treatment, and long-term oncological outcomes of de novo MBC patients undergoing FLC followed by LRT on the primary tumor. The secondary objective was to compare different subgroups of de novo MBC patients to determine the existence of a subset of patients that might benefit from LRT.

## 2. Materials and Methods

### 2.1. Study Design

We retrospectively reviewed all the consecutive de novo MBC patients who underwent FLC followed by LRT and were treated at the Breast Unit of IRCCS Humanitas Research Hospital (Milan, Italy), between October 2006 and January 2020. A multidisciplinary tumor board composed of breast surgeons, breast oncologists, radiotherapists, radiologists, and pathologists discussed the management of each de novo MBC patient. To document the extent of loco-regional disease, all patients underwent pre-operative staging with physical examination and bilateral breast and axillary ultrasound (US). Pre-operative mammography or magnetic resonance imaging (MRI) of the breast was not mandatory. All de novo MBC patients underwent either positron emission tomography (PET) scan or radionuclide bone scan before and after FLC. To confirm the presence of metastatic disease, patients underwent whole-body computed tomography (CT) scan. Diagnosis of invasive breast cancer was histologically confirmed in all patients by core needle biopsy in the breast. If pathological lymph nodes were detected at pre-operative US evaluation, a biopsy of the axilla was performed. A biopsy of the metastatic site was not planned in all de novo MBC patients and the decision to perform it was evaluated individually for each patient. Histopathological confirmation of estrogen receptor (ER) and progesterone receptor (PgR) expression was obtained in all patients using standard immunohistochemical techniques. HER2 status was assessed by immunohistochemistry and defined as negative if the score was 0/1+, equivocal if the score was 2+, or positive if the score was 3+. Equivocal cases were further assessed by fluorescent in situ hybridization, according to the recommendations of the American Society of Clinical Oncology/College of American Pathologists (ASCO/CAP) [20]. The loco-regional and distant tumor response rates were calculated according to Response Evaluation Criteria in Solid Tumors (RECIST) 1.1 criteria [21]. The LRT consisted of complete resection of the primary tumor with either mastectomy or breast-conserving surgery. Regarding axillary staging, sentinel lymph node biopsy (SLNB) was performed in clinically node-negative patients. Axillary lymph node dissection (ALND) was required for patients with a positive lymph node before surgery or with a macrometastatic sentinel lymph node. The exclusion criteria were the following: de novo MBC patients not treated with FLC, metachronous metastatic disease, loco-regional or distant progression of disease during FLC (defined by RECIST criteria), involvement of two visceral organs along with bone metastasis, more than 7 metastatic sites, follow-up ≤ 30 months, lost to follow-up. Each patient provided informed consent for operation and clinical data acquisition.

### 2.2. Stratification Factors and Definitions of Long-Term Oncological Outcomes

Menopausal status was assessed before starting FLC and was defined as pre-menopausal if the patient reported a regular menstrual cycle in her medical history. If the patient gave a history of no menstruation for more than one year she was defined as post-menopausal. The biological subtype of breast cancer was defined as follows: luminal-like (HR+/HER2−), triple-negative (HR−/HER2−), HER2-enriched (HR−/HER2+), triple-positive (HR+/HER2+). Further stratification factors for distant metastases were: site (bone-only, visceral-only, both), and number (1, 2–3, >3). Radiologic complete response (rCR) of the metastatic site was defined as the absence of distant tumor mass at whole-body CT scan or PET uptake after FLC. Progression-free survival (PFS) was defined as the period from the date of LRT to the date of any tumor progression, including loco-regional recurrence or distant metastatic spread. Distant progression-free survival (DPFS) was defined as the time interval from LRT to the date of detection of metastasis to additional organs. Overall survival (OS) was defined as the time interval from LRT to death from any cause or to the last follow-up.

### 2.3. Statistical Analysis

Patients were selected from our prospectively maintained institutional database and retrospectively analyzed. The Kaplan–Meier method with 95% confidence interval (CI) was used to estimate the recurrence and survival probabilities and the log-rank test was used to compare different stratification subgroups of de novo MBC patients according to their menopausal status, biological sub-type, location, number, and rCR of metastases. To determine the sample size for each subgroup of de novo MBC patients, a power analysis was performed. Power was set at 0.8, threshold for significance (α) was set at 0.05, and means and standard deviations of the subgroups were derived from the institutional database. Last follow-up was updated up to September 1, 2022. Statistical significance was set at *p* < 0.05. Data analyses and figures were performed with IBM SPSS 25.0 and SamplePower 3 software (Armonk, NY, USA: IBM Corp).

## 3. Results

### 3.1. Study Population

A total of 61 de novo MBC patients underwent FLC followed by LRT at the Breast Unit of IRCCS Humanitas Research Hospital (Milan, Italy). The median age of the patients was 49 years (range, 30–82 years), and 32 (52.5%) patients were post-menopausal. Bilateral mammography and MRI of the breast were performed in 34 (55.7%) and 19 (31.2%) patients, respectively. The median diameter of the breast tumor before FLC was 38 mm (range, 10–82 mm), and 36 (59.0%) patients were affected by cT2 breast cancer. Regarding FLC treatment protocol: 6 (9.8%) patients received four cycles of anthracycline-based FLC (e.g., AC: 600 mg/m^2^ cyclophosphamide plus 60 mg/m^2^ doxorubicin, administered every 3 weeks), 18 (29.5%) patients received six cycles of anthracycline-based FLC (e.g., EC or AC: 600 mg/m^2^ cyclophosphamide plus 90 mg/m^2^ epirubicin or 60 mg/m^2^ doxorubicin, administered every 3 weeks), and 29 (47.5%) patients received a sequential anthracycline-taxane regimen (e.g., 90 mg/m^2^ epirubicin or 60 mg/m^2^ doxorubicin plus 600 mg/m^2^ cyclophosphamide every 3 weeks for four cycles followed by 75 mg/m^2^ docetaxel for four cycles or by 80 mg/m^2^ paclitaxel for twelve cycles). Sixteen (26.2%) patients received docetaxel (75 mg/m^2^) in combination with anti-HER2 therapies, either Trastuzumab (loading dose of 8 mg/kg followed by 6 mg/kg in subsequent cycles) in twelve patients or Pertuzumab (loading dose of 840 mg followed by 420 mg in subsequent cycles) in the remaining patients, every three weeks. Pre-operative hormone therapy in combination with chemotherapy was administered to 15 (24.6%) patients as part of FLC and it consisted of Tamoxifen and aromatase inhibitors in eight and seven patients, respectively. The majority of de novo MBC patients (54.1%) was affected by luminal-like tumors, followed by triple-negative tumors (21.3%). Overall, 31 (50.8%) patients had visceral-only metastasis, 22 (36.1%) patients had bone-only metastasis, and the rest of the patients had both visceral and bone metastases. Nineteen (31.2%) patients had a single metastasis and twenty (32.8%) patients had more than three metastases. After FLC, 29 (47.5%) patients achieved rCR. The median time from diagnosis to LRT was 7 months (range, 4–27 months). Regarding LRT, the majority of the patients (60.7%) underwent mastectomy. Axillary staging was performed with direct ALND in 35 (57.4%) patients; 20 (32.8%) patients underwent SLNB and 4 of them with subsequent ALND. Patients and tumor baseline characteristics and details on treatment are summarized in Table 1.

### 3.2. Impact of Loco-Regional Treatment on Long-Term Oncological Outcomes and Subgroup Analyses

At a median follow-up of 55 months (range, 32–141 months), 37 (60.7%) patients underwent loco-regional and/or distant progression of disease and 30 (49.2%) patients died. Overall, 2 patients had loco-regional recurrence, 26 patients experienced development of new metastases, and 9 patients had both loco-regional and distant recurrence. Loco-regional recurrences were treated with mastectomy, ALND, or surgical scar removal. New metastases were treated with a combination of chemotherapy (e.g., doxorubicin, capecitabine, paclitaxel, lapatinib, palbociclib, abemaciclib, eribulin, vinorelbine, and fulvestrant), hormone therapy, Trastuzumab, bone and whole-brain radiotherapy, and bone or brain surgical resection. Long-term oncological outcomes of all patients with de novo MBC were analyzed. The median PFS, DPFS, and OS were 36 months (range, 2–136 months), 39 months (range, 2–136 months), and 54 months (range, 3–140 months), respectively. The PFS, DPFS, and OS rates at 5 years were 40.9% (95% CI 17.0–61.2), 41.9% (95% CI 20.2–59.3), and 58.8% (95% CI 56.4–148.6), respectively. We then evaluated the effect of LRT in different subgroups of de novo MBC patients divided by: menopausal status, biological sub-type, location, number, and rCR of metastases (Table 2).

There was no significant difference in terms of PFS, DPFS, and OS between different subgroups of de novo MBC patients according to menopausal status (pre-menopausal versus post-menopausal; *p* = 0.591, *p* = 0.540, *p* = 0.153; respectively, Figure 1), location of metastases (visceral-only versus bone-only versus visceral and bone; *p* = 0.115, *p* = 0.172, *p* = 0.723; respectively, Figure 2), number of metastases (1 versus 2–3 versus >3, *p* = 0.075, *p* = 0.119, *p* = 0.283; respectively, Figure 3), and radiologic response after FLC (rCR versus no rCR, *p* = 0.474, *p* = 0.522, *p* = 0.081; respectively, Figure 4). A trend for better PFS and DPFS was achieved for triple-positive tumors compared with luminal-like, triple-negative, and HER2-enriched sub-types (*p* = 0.058, *p* = 0.069; respectively, Figure 5); however, no statistical significance was achieved with respect to OS (*p* = 0.625, Figure 5).

## 4. Discussion

In this small retrospective cohort analysis, we sought to disclose the long-term oncological outcomes of de novo MBC patients undergoing FLC followed by LRT and to determine the potential existence of a subgroup of metastatic patients who might benefit from surgery. Our results suggest that there is no statistically significant survival advantage in any subgroup of de novo MBC patients, regardless of their menopausal status, surgical treatment, biological sub-type, location, number, and rCR of metastases. However, we observed a slight trend toward better recurrence outcomes in triple-positive tumors.

Stage IV breast cancer is considered an incurable systemic disease and survival is mainly determined by the progression of metastases [22]. Until now, systemic therapy, including chemotherapy, anti-HER2-targeted therapy, and/or endocrine therapy, remains the mainstay of treatment. The role of LRT remains controversial and is usually reserved only for patients with impending complications, such as skin ulceration, bleeding, fungation, and pain [4,23]. Despite these recommendations, the benefit of LRT in this population has long been debated and a vast number of MBC patients still undergo surgical resection of the primary tumor [14,22]. Khan et al. [24] examined the use of local therapy and its impact on survival in 16,023 patients with stage IV breast cancer in an analysis of the National Cancer Database from 1990 to 1993. The majority (57.2%) of these patients underwent partial or total mastectomy. The observed 3-year OS rate was 24.9%, 17.3% without surgery, 27.7% with segmental mastectomy, and 31.8% with total mastectomy (*p* < 0.001). The presence of free surgical margins was associated with an improvement in 3-year survival, regardless of the type of surgery. Gnerlich et al. [25] conducted a retrospective, population-based cohort study of 9734 stage IV breast cancer patients using data from the Surveillance, Epidemiology, and End Results (SEER) program data from 1998 to 2003 and found a longer median survival for women who underwent surgery compared with women who did not. These findings were also supported by an analysis of 300 MBC patients recorded at the Geneva Cancer Registry between 1977 and 1996 [26], in which the recorded 5-year OS was 16% (27% with negative margins, 16% with positive margins, and 12% without surgery). In a more recent study, Vohra et al. [27] analyzed outcomes of 29,916 MBC patients from the SEER database (1988–2011), proposing a survival advantage with surgical intervention (median OS 34 months for surgery versus 18 months without surgery). However, data on HER2 status were incomplete and no stratified analysis was conducted. In contrast, Babiera et al. [28] examined the records of 224 patients with stage IV disease treated at the University of Texas MD Anderson Cancer Center between 1997 and 2002. Eighty-two patients (37%) were treated with surgery of the primary tumor (48% with segmental mastectomy and 52% with mastectomy) and 142 (63%) were treated with systemic therapy alone. At a median follow-up of 32.1 months, no improvement in OS was observed. Patients who underwent surgery were more likely to have bone and liver disease as compared with brain and lung metastases and they were more likely to have HER2-enriched breast cancer. Moreover, Dominici et al. [29] performed a retrospective matched analysis of the NCCN Breast Cancer Outcomes Database from 1997 to 2007 of 551 de novo MBC patients and found that survival was similar between the patients treated with surgery and without surgery (3.5 years versus 3.4 years, respectively). In addition, a similar matched-pair analysis [30], performed on 622 stage IV breast cancer patients, found no difference in survival between patients treated with or without surgery, suggesting that case selection bias of LRT-group patients may explain most, if not all, of the apparent survival benefit.

Previous retrospective studies reported contrasting results with LRT of the primary tumor in the setting of stage IV disease. Nevertheless, due to selection bias that may have affected the results, MBC patients selected for LRT may have been healthier and/or had a lower metastatic disease burden. In fact, various factors, including: age, comorbidities, performance status, and overall organ function play a fundamental role in the heterogeneity of the MBC population [31]. The potential impact of selection bias and heterogeneity was partially overcome by the matched-analysis cohorts, which did not show better survival outcomes. With this in mind, prospective trials were designed to determine the impact of LRT on the prognosis of MBC patients. The Translational Breast Cancer Research Consortium 013 (TBCRC 013) [32] is a multicenter prospective registry study evaluating the role of surgery for the primary tumor in de novo stage IV disease. From 2009 to 2012, 128 patients with stage IV disease at presentation or stage IV within three months of diagnosis were enrolled. All patients received FLC, and patients classified as responders were referred for discussion of elective surgery. The majority (85%) of patients responded to FLC and 3-year OS was superior among responders than non-responders (*p* < 0.001). However, among responders, surgery did not impact OS, regardless of tumor sub-type. These major findings suggest that improved long-term oncological outcomes may be due to response to FLC rather than LRT; therefore, caution is suggested in selecting patients for LRT outside of clinical trials. The ABCSG-28 POSYTIVE trial [33] was a prospective, randomized, phase III study comparing the median survival between previously untreated de novo MBC patients undergoing primary surgery followed by systemic therapy (Arm A) or primary systemic therapy alone (Arm B). The trial was stopped early due to poor recruitment; however, 90 well-balanced patients (45 arm A, 45 arm B) were included in the study, with a median follow-up of 37.5 months. Median survival in arm A was 34.6 months versus 54.8 months in the non-surgery arm (*p* = 0.267), and time to distant progression was 13.9 months in the surgery arm versus 29.0 months in the non-surgery arm (*p* = 0.0668); no prognostic advantage was demonstrated for surgical resection of the primary tumor in de novo MBC patients. Soran et al. [19] performed a prospective, multicenter, randomized, phase III controlled trial (MF07-01) focused on the impact of breast surgery on survival in de novo MBC patients. Patients were randomized 1:1, with one group receiving sequential systemic therapy after LRT and the other group receiving systemic therapy alone. A total of 274 patients were enrolled, and after a median follow-up of 3 years, there was no survival benefit for the LRT group. An unplanned subgroup analysis showed that patients with HR+ or HER2− breast cancer, solitary bone metastasis, and younger than 55 years had a significant survival benefit from initial surgery; however, there was no differential improvement in survival for patients with triple-negative breast cancer. Badwe et al. [18] performed an open-label randomized controlled trial of 350 previously untreated de novo MBC patients between 2005 and 2013. The median OS was 19.2 months in the surgical group versus 20.5 months in the non-surgical group (*p* = 0.79), demonstrating that LRT does not provide a survival benefit in the de novo MBC population. Additionally, there was no evidence that LRT derived any prognostic benefit for any patient subgroup identified by menopausal status, metastatic disease burden, HR, and HER2 status.

As highlighted by previous clinical trials, the management of MBC remains a challenge, requiring a complex decision-making process regarding who might be an appropriate candidate for LRT. Interestingly, the analyses of the clinical and metabolic phenotypes, including the modulation of miRNAs and adipokines regulating metabolism [34], and the molecular links between thyroid autoimmunity and breast cancer [35], may lead to the identification of targeted approaches to treat MBC.

The major limitation of our study is represented by the small number of de novo MBC patients included; in fact, no subgroup reached the optimal sample size for analysis. Moreover, the retrospective approach and the potential selection bias may have affected the analysis. However, our study also presents some strong points. Clear inclusion and exclusion criteria allowed us to select a homogeneous, well-balanced population of de novo MBC patients, the observation period is relatively long, and no patient was lost to follow-up.

## 5. Conclusions

In conclusion, no specific subgroup of de novo MBC patients showed a statistically significant survival advantage after FLC followed by LRT of the primary tumor, irrespective of their menopausal status, biological sub-type, location, number, and radiologic response of metastases. Nevertheless, in a small subgroup of triple-positive patients treated with surgery, there was a slight trend toward better PFS and DPFS compared with the other subtypes. Our study highlights the importance of pre- and post-operative systemic treatment with endocrine and anti-HER2 targeted therapy in MBC patients; however, larger randomized prospective clinical trials are needed to confirm our results.

## Figures and Tables

**Figure 1 cancers-14-06237-f001:**
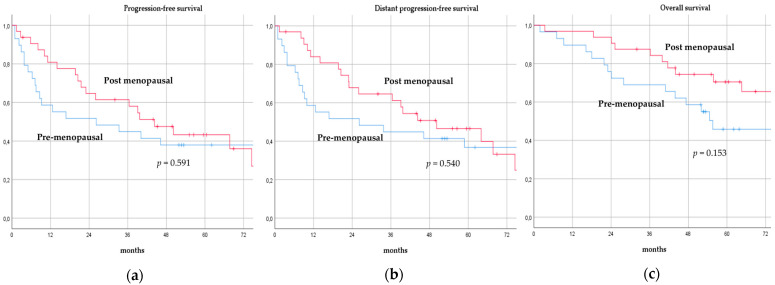
Progression-free survival (**a**), distant progression-free survival (**b**), and overall survival (**c**) curves of de novo metastatic breast cancer patients according to menopausal status.

**Figure 2 cancers-14-06237-f002:**
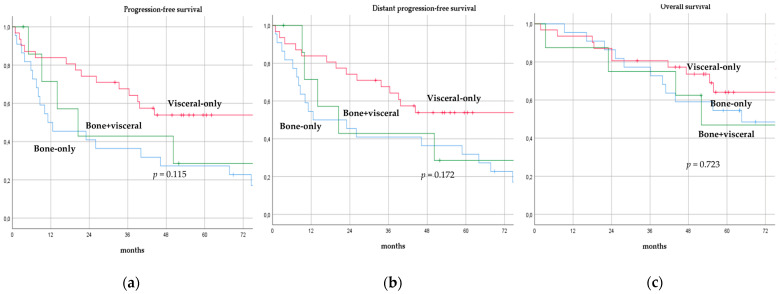
Progression-free survival (**a**), distant progression-free survival (**b**), and overall survival (**c**) curves of de novo metastatic breast cancer patients according to location of metastases.

**Figure 3 cancers-14-06237-f003:**
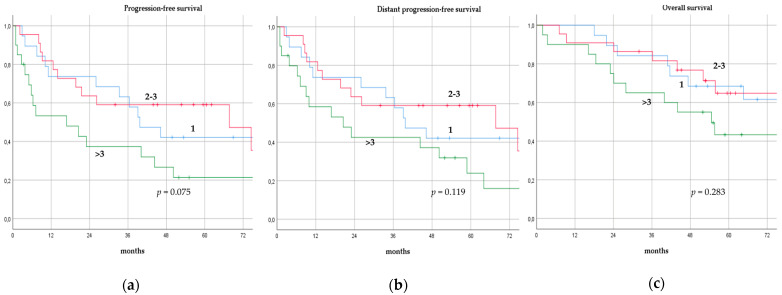
Progression-free survival (**a**), distant progression-free survival (**b**), and overall survival (**c**) curves of de novo metastatic breast cancer patients according to number of metastases.

**Figure 4 cancers-14-06237-f004:**
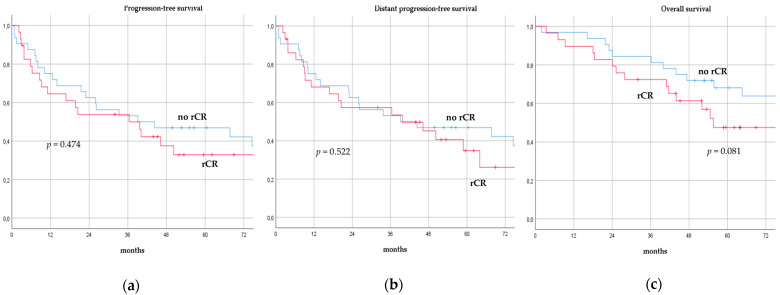
Progression-free survival (**a**), distant progression-free survival (**b**), and overall survival (**c**) curves of de novo metastatic breast cancer patients according to radiologic response after front-line chemotherapy. rCR: Radiologic complete response.

**Figure 5 cancers-14-06237-f005:**
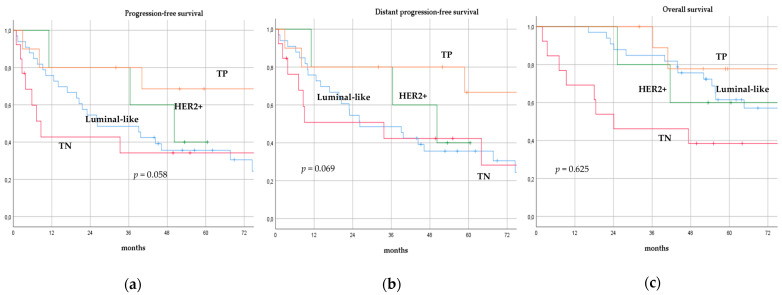
Progression-free survival (**a**), distant progression-free survival (**b**), and overall survival (**c**) curves of de novo metastatic breast cancer patients according to biological sub-type. TP: Triple-positive, HER2+: HER2-enriched, TN: Triple-negative.

**Table 1 cancers-14-06237-t001:** Characteristics and treatment of 61 de novo metastatic breast cancer patients undergoing front-line chemotherapy and loco-regional treatment.

Characteristics	Number (%)/Median (Range)
**Patients**	
Age (years)	49 (30–82)
Post-menopausal	32 (52.5%)
**Radiological staging**	
Mammography	34 (55.7%)
Breast and axillary US	61 (100%)
MRI	19 (31.2%)
PET	49 (80.3%)
Radionuclide bone scan	16 (26.2%)
**Pre-operative treatment**	
- Anthracycline only	24 (39.3%)
- Anthracycline and taxanes	29 (47.5%)
- Trastuzumab	16 (26.2%)
- Pertuzumab	4 (6.6%)
- Hormone therapy	15 (24.6%)
**Breast cancer**	
Single nodule	46 (75.4%)
Dimension pre-FLC (mm)	38 (10–82)
Stage pre-FLC	
- cT1	8 (13.1%)
- cT2	36 (59.0%)
- cT3	7 (11.5%)
- cT4	10 (16.4%)
- cN0	7 (11.5%)
- cN1	54 (88.5%)
Histotype	
- Ductal	57 (93.4%)
- Lobular	3 (4.9%)
- Mucinous	1 (1.7%)
Stage post-FLC	
- ypT0	6 (9.8%)
- ypTis	3 (4.9%)
- ypT1a	4 (6.6%)
- ypT1b	6 (9.8%)
- ypT1c	11 (18.0%)
- ypT2	19 (31.2%)
- ypT3-4	12 (19.7%)
- Nx	6 (9.8%)
- ypN0	22 (36.1%)
- ypN1a	12 (19.7%)
- ypN2	13 (21.3%)
- ypN3	8 (13.1%)
Dimension post-FLC (mm)	21 (0–81)
Biological sub-types	
- Luminal-like	33 (54.1%)
- Triple-negative	13 (21.3%)
- HER2-enriched	5 (8.2%)
- Triple-positive	10 (16.4%)
Ki67 (%)	14 (2–90)
Lymphovascular invasion	20 (32.8%)
**Metastasis**	
Radiologic complete response	29 (47.5%)
Location	
- Visceral	31 (50.8%)
- Bone	22 (36.1%)
- Visceral and bone	8 (13.1%)
Number	
- 1	19 (31.2%)
- 2–3	22 (36.1%)
- >3	20 (32.8%)
**Loco-regional treatment**	
Surgery	
- BCS	24 (39.3%)
- Mastectomy	37 (60.7%)
- ALND	39 (63.9%)
**Post-operative treatment**	
- Loco-regional radiotherapy	42 (68.9%)
- Hormone therapy	43 (70.5%)
- Chemotherapy	9 (14.8%)
- Trastuzumab	15 (24.6%)

US: Ultrasonography, MRI: Magnetic resonance imaging, PET: Positron emission tomography, FLC: Front-line chemotherapy, HER2: HER2 evaluated either on immunohistochemistry or on in-situ hybridization, according to the ASCO CAP guidelines, BCS: Breast-conserving surgery, ALND: Axillary lymph node dissection.

**Table 2 cancers-14-06237-t002:** Comparison of progression-free, distant progression-free, and overall survival rates of different subgroups of patients with de novo metastatic breast cancer.

Subgroups	5-Year PFS (95% CI)	5-Year DPFS (95% CI)	5-Year OS (95% CI)
Menopausal status			
- Pre-menopausal	37.9% (17.0–62.6)	36.8% (20.2–63.3)	45.8% (26.7–84.5)
- Post menopausal	43.2% (27.6–61.0)	46.5% (24.9–75.5)	70.5% (64.2–140.8)
Biological sub-types			
- Luminal-like	35.6% (5.7–46.7)	35.6% (7.3–45.1)	61.5% (47.7–157.3)
- Triple-negative	34.2% (4.0–43.2)	42.3% (20.2–75.2)	38.5% (36.4–58.5)
- HER2-enriched	40.0% (20.6–79.8)	40.0% (20.6–79.8)	60.0% (56.4–148.5)
- Triple-positive	68.6% (17.0–71.2)	66.7% (20.2–69.3)	77.8% (59.4–158.5)
Location			
- Visceral	53.9% (17.0–61.2)	53.9% (20.2–59.3)	64.1% (17.7–203.5)
- Bone	27.3% (17.0–27.7)	31.8% (20.2–40.9)	54.5% (17.3–111.9)
- Visceral and bone	28.6% (3.8–61.2)	28.6% (3.8–37.2)	46.9% (15.9–148.6)
Number			
- 1	42.1% (25.8–53.6)	42.1% (25.8–53.6)	68.4% (48.2–156.9)
- 2–3	59.1% (13.7–121.7)	59.1% (13.7–121.7)	64.8% (56.4–148.6)
- >3	21.3% (17.0–61.2)	23.9% (2.0–39.0)	43.3% (32.6–76.6)
Radiologic response after FLC			
- rCR	32.9% (11.3–61.5)	34.8% (6.4–73.0)	47.5% (34.2–77.0)
- no rCR	46.9% (17.0–90.2)	46.9% (20.2–90.2)	68.1% (65.6–206.8)

PFS: Progression-free survival, DPFS: Distant progression-free survival, OS: Overall survival, CI: Confidence interval, FLC: Front-line chemotherapy, rCR: Radiologic complete response.

## Data Availability

The data presented in this study are available within the article.

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
