# Peer review of "Loco-Regional Treatment of the Primary Tumor in De Novo Metastatic Breast Cancer Patients Undergoing Front-Line Chemotherapy"

_cancers, 2022, doi:10.3390/cancers14246237_

Round 1
Reviewer 1 Report
The Authors undertook important topic in current oncology. They tried to assess the role of local therapy in breast cancer patients with distant metastases.
1. The Authors should add more information about group of patients (time from diagnosis of MBC to operation; explenation of hormone therapy - the Authors stated that all patients were treated upfront with chemotherapy, but then claimed that hormone therapy was used in 15 patients - 24,6%).
2. Please add medians of PFS, DPFS and OS - there is only information about % of patients at 5 years, but in many analyses median was reached.
3. The authors must change the “Discussion” section and the title of article. All analyses and results do not support the statement that surgery does not improve results. Actually, the results indicate that the outcome was impressive - for example 5-year mPFS in TNBC was close to 34% (high). In clinical trials with new drugs mOS is close to 2 years, it suggests that surgery was relevant. In the presented study the Authors did not compare the surgery population with patients with only systemic therapy - so it is impossible to state that surgery had no effect. The authors can only suggest that there was no difference in results in population with surgery according to menopausal status, location of metastases, number of metastases, radiologic response aftre front-line chemotherapy as well as biological subtype.
Overall, the manuscript is generally well-written and the topic is important and appropriate for Cancers and its audience. It gives the reader interesting information related to the patients with de novo stage IV BC. The manuscript can be published after the above mentioned changes.
Author Response
POINT-BY-POINT REPLY TO REVIEWERS’ COMMENTS
We thank the editors and reviewers of CANCERS for the opportunity to reply to reviewers’ comments. We do believe that now the manuscript is much more precise and will be of interest to the readers of CANCERS. Please find our replies in this document in bold black.
Reviewers' Comments & Replies:
Reviewer #1: The Authors undertook important topic in current oncology. They tried to assess the role of local therapy in breast cancer patients with distant metastases.
- The Authors should add more information about group of patients (time from diagnosis of MBC to operation; explenation of hormone therapy - the Authors stated that all patients were treated upfront with chemotherapy, but then claimed that hormone therapy was used in 15 patients - 24,6%).
Reply: We thank the reviewer for the comment.
The median time from diagnosis to LRT was added in the Results section.
In 15 patients pre-operative hormone therapy was used in combination with chemotherapy as part of FLC. This was added in the Results section for clarification.
- Please add medians of PFS, DPFS and OS - there is only information about % of patients at 5 years, but in many analyses median was reached.
Reply: We thank the reviewer for the comment.
The median PFS, DFPS, and OS were added in the Results section.
- The authors must change the “Discussion” section and the title of article. All analyses and results do not support the statement that surgery does not improve results. Actually, the results indicate that the outcome was impressive - for example 5-year mPFS in TNBC was close to 34% (high). In clinical trials with new drugs mOS is close to 2 years, it suggests that surgery was relevant. In the presented study the Authors did not compare the surgery population with patients with only systemic therapy - so it is impossible to state that surgery had no effect. The authors can only suggest that there was no difference in results in population with surgery according to menopausal status, location of metastases, number of metastases, radiologic response aftre front-line chemotherapy as well as biological subtype.
Reply: We thank the reviewer for the comment.
The Title and the Discussion section were modified accordingly.
Overall, the manuscript is generally well-written and the topic is important and appropriate for Cancers and its audience. It gives the reader interesting information related to the patients with de novo stage IV BC. The manuscript can be published after the above mentioned changes.

Reviewer 2 Report
I read the manuscript with interest. Management of metastatic breast cancer remains a challenge for clinicians, and there is debate about the evidence suggesting that locoregional treatment of the primary tumor confers an overall survival advantage in this setting. Metastatic breast cancer represents a disease characterized by tremendous heterogeneity, as described by many authors. I appreciate the effort of the authors.
My main comments are as follows:
- There are gaps in knowledge that may impact decision-making regarding who is a good candidate for locoregional resection in MBC. Specifically, health status, i.e., age, comorbidities, performance status, and organ function, contribute to MBC presentation, influencing treatment decisions, and patient outcomes. In this light, what risk factors need to be identified and then treated to improve the prognosis of patients with MBC remains unclear. Although clinically evident metastasis is usually associated with advanced stages of cancer development, micrometastatic dissemination may be an early phenomenon. Inconclusive data are available on molecular events, including changes in specific metabolic pathways underlying the development of metastatic disease, and this may influence treatment decision-making and, in part, may influence the response to locoregional surgical treatment. Indeed, some questions remain unanswered: what risk factors related to MBC prognosis need to be identified? Do metabolic changes affect the outcome of the surgical MBC procedure? Are specific metabolic interventions available in this context? The patient's overall health condition, which affects cancer progression and morbidity and their associated molecular targets, must be considered the in this paper. In the present article, one must also focus on both locoregional surgical strategies in MBC and whether concomitant metabolic derangements may play a role in prognosis. Metabolic syndrome is often associated with the imbalance of hormones and adipokines or the impact of thyroid axis's function on BC progression. The authors are advised to expand the bibliography.
- Both the goals and conclusions of this revision are broad and radical
- Clarify all abbreviations used in the text (Line 68).
- I consider it interesting after this revision.
Author Response
POINT-BY-POINT REPLY TO REVIEWERS’ COMMENTS
We thank the editors and reviewers of CANCERS for the opportunity to reply to reviewers’ comments. We do believe that now the manuscript is much more precise and will be of interest to the readers of CANCERS. Please find our replies in this document in bold black.
Reviewers' Comments & Replies:
Reviewer #2: I read the manuscript with interest. Management of metastatic breast cancer remains a challenge for clinicians, and there is debate about the evidence suggesting that locoregional treatment of the primary tumor confers an overall survival advantage in this setting. Metastatic breast cancer represents a disease characterized by tremendous heterogeneity, as described by many authors. I appreciate the effort of the authors.
My main comments are as follows:
- There are gaps in knowledge that may impact decision-making regarding who is a good candidate for locoregional resection in MBC. Specifically, health status, i.e., age, comorbidities, performance status, and organ function, contribute to MBC presentation, influencing treatment decisions, and patient outcomes. In this light, what risk factors need to be identified and then treated to improve the prognosis of patients with MBC remains unclear. Although clinically evident metastasis is usually associated with advanced stages of cancer development, micrometastatic dissemination may be an early phenomenon. Inconclusive data are available on molecular events, including changes in specific metabolic pathways underlying the development of metastatic disease, and this may influence treatment decision-making and, in part, may influence the response to locoregional surgical treatment. Indeed, some questions remain unanswered: what risk factors related to MBC prognosis need to be identified? Do metabolic changes affect the outcome of the surgical MBC procedure? Are specific metabolic interventions available in this context? The patient's overall health condition, which affects cancer progression and morbidity and their associated molecular targets, must be considered the in this paper. In the present article, one must also focus on both locoregional surgical strategies in MBC and whether concomitant metabolic derangements may play a role in prognosis. Metabolic syndrome is often associated with the imbalance of hormones and adipokines or the impact of thyroid axis's function on BC progression. The authors are advised to expand the bibliography.
- Both the goals and conclusions of this revision are broad and radical
Reply: We thank the reviewer for the comment.
The Discussion section was modified accordingly.
- Clarify all abbreviations used in the text (Line 68).
Reply: We thank the reviewer for the comment.
It was clarified.
- I consider it interesting after this revision.

Round 2
Reviewer 1 Report
Thank you to consider my comments. Article can be published in this version.